# A Parsimonious Description and Cross-Country Analysis of COVID-19 Epidemic Curves

**DOI:** 10.3390/ijerph17186487

**Published:** 2020-09-06

**Authors:** Kristoffer Rypdal, Martin Rypdal

**Affiliations:** Department of Mathematics and Statistics, UiT—The Arctic University of Norway, 9019 Tromso, Norway; martin.rypdal@uit.no

**Keywords:** COVID-19, epidemic curve, logistic curve, Gompertz model, death toll

## Abstract

In a given country, the cumulative death toll of the first wave of the COVID-19 epidemic follows a sigmoid curve as a function of time. In most cases, the curve is well described by the Gompertz function, which is characterized by two essential parameters, the initial growth rate and the decay rate as the first epidemic wave subsides. These parameters are determined by socioeconomic factors and the countermeasures to halt the epidemic. The Gompertz model implies that the total death toll depends exponentially, and hence very sensitively, on the ratio between these rates. The remarkably different epidemic curves for the first epidemic wave in Sweden and Norway and many other countries are classified and discussed in this framework, and their usefulness for the planning of mitigation strategies is discussed.

## 1. Introduction

At the time of writing, the COVID-19 pandemic is expanding exponentially in many countries, particularly in the tropics and the southern hemisphere. On the other hand, in Southeast Asia, Europe, and North America, the first wave of infection is mostly over. In some countries, e.g., the United States, Israel, Iran, and Spain, a second and stronger wave develops. For the first wave, the general shape of the epidemic curves, the daily number of confirmed cases or COVID-19 related deaths, is one of rapid growth followed by a slower decay. However, even though this general characteristic is ubiquitous, the total death toll per million inhabitants in comparable countries varies by more than an order of magnitude. For instance, by 5 August 2020, Sweden had registered 569 deaths per million, while neighboring Norway only 47.

The debate about the causes of this pronounced variability between countries has involved a plethora of explanations [1] based on correlation techniques such as multiple regression [2], or on models that involve statistical modelling of the dynamics of transmission [3]. Suggested predictors (explaining variables) in such analyses have encompassed demographic, socioeconomic, and health-related characteristics, and strength and timing of border closures and social distancing measures. Some success in assessing the effect of non-pharmaceutical interventions in 11 European countries was obtained in [3], and the effect of social distancing measures in Sweden, Denmark and Norway was studied by a considerably simpler approach in [4]. The three countries are similar in terms of health care, language, culture, climate, economy, and institutional framework. Moreover, due to the simultaneous import of infected cases via ski tourists returning from Austria and Italy in early March 2020, it is plausible that community spread of the disease started at approximately the same time in these countries. On the other hand, while Norway and Denmark followed the general lockdown of most of the European Union countries on 12 March 2020, Sweden followed a different path, largely using voluntary measures [5].

On a world scale, cross-country comparative analysis is extremely challenging for at least two reasons. One is that the relevant predictors vary across the world and may attain an unmanageable number. Another is that the same problem could pertain to the predictands (explained variable). How do we best describe the impacts of the disease in a way that allows comparison between countries in different parts of the world? Some studies employ regression analysis to a sample containing the majority of the world’s countries, but consider the accumulated COVID-19 cases and/or deaths up to a certain date as predictand [6]. This variable may be very misleading, however, since the virus was not introduced simultaneously in all countries. In principle, we need to predict and compare the full epidemic curves across countries, but in order to extract information from from such comparison, we need a simplified characterization of these curves, i.e., we need to reduce the information in the curves to a few numbers. This paper is about how we can obtain and make use of such a characterization.

Our main finding is that the epidemic curves for COVID-19 related deaths for most countries with a reliable reporting system are surprisingly well described by the so-called Gompertz growth model [7]. This model contains only two essential free parameters to be determined by the data. These parameters determine the initial relative growth rate of the number of daily deaths and the decay rate of this number as the first epidemic wave comes to an end. The mathematical properties of the Gompertz model imply that the total death toll in the first wave is given by the exponential of the ratio between these two rates. Countries with rapid initial growth and slow later decay suffer the higher death toll, but there are variations among countries with a similar number of deaths. For instance, while Spain experienced very fast initial growth followed by a rapid decay due to a very forceful lockdown, Swedish death numbers rose and decayed considerably more slowly. Other countries, like Norway, experienced slower initial growth than Sweden, and faster decay, and thereby obtained death numbers one order of magnitude lower. Since the initial growth rate in Norway was so much lower than in Spain, Norway could obtain these results with slower decay, i.e., with a considerably milder lockdown, than Spain.

In this paper, we make a detailed comparison of epidemic curves for Sweden and Norway, including computation and plotting the respective time evolution of the relative growth rates and time-dependent reproduction numbers. We further fit the Gompertz model to the epidemic curves (deaths) for a large number of countries that have completed the first wave, and plot the model parameters in a diagram highlighting their differences. The two rates are predictands that correlate in different ways to various predictors of the epidemic curve, and via these predictands they determine the total death toll in the first wave. Regressions that determine the coefficients of these predictors will be useful in the design of optimal mitigation strategies for upcoming waves of the epidemic.

## 2. Materials and Methods

### 2.1. Data

Data employed in this paper are downloaded from Our World in Data [8]. Their data for COVID-19 related deaths are retrieved from the European Centre for Disease Prevention and Control. As explained by the COVID-19 Health System Response Monitor [9], figures may vary among countries and may complicate cross-country analysis. This problem is most serious for the headline figures presenting the most recent day-to-day data. We use data published at the end of July, but only data up to the first week of July. In Section 4 we will discuss the implications of such uncertainties for the conclusions of the analysis.

The official figures from China suffer from 40% discontinuous increase in death numbers on 17 April. The official explanation is that home deaths before that date were included. In our analysis, we account for this by adjusting the numbers before 17 April by a factor of 1.4, thus removing the discontinuity.

### 2.2. The Gompertz Model

The Gompertz model is a special case of the class of logistic growth models [10]. This general class are solutions J(t) to the nonlinear, separable differential equation
(1)$$\frac{dJ}{dt}=\gamma \left(J\right)\;J,$$
where *t* is the time coordinate and *J* is a quantity undergoing monotonic growth that saturates at the carrying capacity $J_{\infty }$ in the limit t→∞. In the Gompertz model the instantaneous relative growth rate $\gamma \left(J\right)=d_{t}J/J=d_{t}\left(\ln J\right)$ is (see Appendix A),
(2)$$\gamma \left(J\right)=\gamma_{\infty }\ln\left(\frac{J_{\infty }}{J}\right),$$
and Equation (Equation 1) can be integrated to yield
(3)$$\ln J(t)=\ln J_{\infty }-\ln\left(\frac{J_{\infty }}{J_{0}}\right)\;\exp\left(-\gamma_{\infty }t\right),$$
or equivalently,
(4)$$J\left(t\right)=J_{\infty }{\left(\frac{J_{0}}{J_{\infty }}\right)}^{\exp\left(-\gamma_{\infty }t\right)}.$$

Note that in this limit γ(J) diverges as J→0, so Equation (Equation 1) should be understood as an initial value problem with $J\left(0\right)=J_{0}$ and an initial growth rate
(5)$$\gamma_{0}=\gamma_{\infty }\ln\left(\frac{J_{\infty }}{J_{0}}\right).$$

The logarithmic form of Equation (Equation 3) shows that the Gompertz model describes a growth where the logarithm of *J* converges exponentially towards the limit $\ln J_{\infty }$, i.e., the difference $J_{\infty }-J$ decays with the decay rate $\gamma_{\infty }$. The growth curve is completely determined by the initial value $J_{0}$, the carrying capacity $J_{\infty }$, and the asymptotic decay rate $\gamma_{\infty }$. These features make it natural to plot J(t) in a logarithmic plot.

From Equation (Equation 3) we have $\ln\left(J_{\infty }/J\right)=\exp\left(-\gamma_{\infty }t\right)\ln\left(J_{\infty }/J_{0}\right)$, and Equation (Equation 2) then yields γ as a function of time,
(6)$$\gamma \left(t\right)=\gamma_{0}\exp\left(-\gamma_{\infty }t\right).$$

This is an alternative way of expressing Equations (Equation 1) and (Equation 2), which shows that the relative growth rate as a function of time decays exponentially with rate $\gamma_{\infty }$ for all t>0 and is a remarkable result since it implies that γ(t) depends only on $\gamma_{0}$ and $\gamma_{\infty }$. Equation (Equation 5) shows that $\gamma_{0}$ is determined by the three model parameters $J_{0}$, $J_{\infty }$, and $\gamma_{\infty }$, and hence can replace any of these as a model parameter. The parameters $J_{\infty }$, and $\gamma_{\infty }$ are more fundamental than the others, however, since the latter depend on the choice of the time origin. Equation (Equation 4) implies that a translation of the time variable, $t\to t-t_{0}$, corresponding to a shift of the time origin, leads to a shift of the initial value; $J_{0}\to J_{\infty }{\left(J_{0}/J_{\infty }\right)}^{\exp\left(-\gamma_{\infty }t_{0}\right)}$, and Equation (Equation 6) leads to a shift of the initial growth rate; $\gamma_{0}\to \gamma_{0}\exp\left(\gamma_{\infty }t_{0}\right)$. A natural choice is to choose the time origin to be the first day the observed value of *J* exceeds one death per million nhabitants.

In the fitting procedure we do not fix $J_{0}=1$, thus allowing the modeled value $J_{0}$ to be slightly different from one and from the observed value at t=0. Equation (Equation 5) can now be rewritten to give us the total death toll in terms of the rates $\gamma_{0}$ and $\gamma_{\infty }$,
(7)$$J_{\infty }=J_{0}\exp\left(\frac{\gamma_{0}}{\gamma_{\infty }}\right),$$
where $J_{0}$ is close to unity. The significance of Equation (Equation 7) is that the accumulated death toll over the first epidemic wave depends exponentially on the ratio of the growth rate $\gamma_{0}$ at the time when the number of deaths exceeds one per million, to the asymptotic decay rate $\gamma_{\infty }$ as the epidemic burns out. A more asymmetric epidemic curve, rapid rise and slow decay, intuitively leads to a larger number of deaths, but the exponential dependence on the ratio of rates suggests that the sensitivity of the death toll with respect to countermeasures is surprisingly high.

The monotonically increasing sigmoid shape of J(t) suggests that it models the evolution of accumulated quantities like the cumulative number of infected individuals or cumulative deaths. In the literature one also frequently deals with the daily numbers, i.e. with the derivative $d_{t}J$. Figure 1a shows an example of J(t) given by the Gompertz function, and Figure 1b its derivative, both in logarithmic plots. In Figure 1b
(8)$$\Gamma \left(t\right)=d_{t}\ln\left(d_{t}J\right)=\gamma_{\infty }\left(e^{-\gamma_{\infty }t}\ln\left(\frac{J_{\infty }}{J_{0}}\right)-1\right)=\gamma_{\infty }\left[\ln\left(\frac{J_{\infty }}{J(t)}\right)-1\right].$$
is the relative growth rate of the daily number $d_{t}J$ which is a skew, bell-shaped function. Γ(t) is a monotonically decreasing function starting out at the positive value $\Gamma_{0}=\gamma_{\infty }(\ln\left(J_{\infty }/J_{0})-1\right)$, crossing zero at the peak of the $d_{t}J$-curve, and converging towards the negative value $-\gamma_{\infty }$ as t→∞. From Equation (Equation 8) we find an alternative to Equation (Equation 7)
(9)$$J_{\infty }=J_{0}\exp\left(1+\frac{\Gamma_{0}}{\gamma_{\infty }}\right)=\left(J_{0}e\right)\exp\left(\frac{\Gamma_{0}}{\gamma_{\infty }}\right),$$
and it follows that
(10)$$\Gamma_{0}=\gamma_{0}-\gamma_{\infty }.$$

Thus, we observe that the total death toll depends exponentially on the ratio $\Gamma_{0}/\gamma_{\infty }$, which is the absolute value of the ratio of the initial and the final slopes of the curve in Figure 1b.

The parameters $J_{\infty }$ and $\gamma_{\infty }$ are intrinsic in the sense that they do not depend on the time origin, while $\gamma_{0}$ and $\Gamma_{0}$ do exhibit such a dependence. This is reflected by the inconvenient fact that the estimated $J_{0}$ is different for each country and is determined by the choice we have made for the time origin (the first day the number of deaths per million exceeds one). For comparison of countries, it would be more correct to use the growth rate $\gamma_{1}$ at the time $t_{1}$ when $J(t_{1})=1$. This growth rate is found by putting J=1 in Equations (Equation 2) and (Equation 8), yielding
(11)$$\gamma_{1}=\gamma_{\infty }\ln J_{\infty },\;\;\;\Gamma_{1}=\gamma_{1}-\gamma_{\infty },$$
such that Equations (Equation 7) and (Equation 9) reduce to,
(12)$$J_{\infty }=\exp\left(\frac{\gamma_{1}}{\gamma_{\infty }}\right)=\exp\left(1+\frac{\Gamma_{1}}{\gamma_{\infty }}\right)=e\exp\left(\frac{\Gamma_{1}}{\gamma_{\infty }}\right),$$
or, by taking the logarithm,
(13)$$\ln J_{\infty }=\frac{\gamma_{1}}{\gamma_{\infty }}=1+\frac{\Gamma_{1}}{\gamma_{\infty }}$$

### 2.3. The Evolution of the Reproduction Number

In Appendix A the Gompertz model is derived heuristically from the SIR dynamical model with some auxiliary assumptions. In the SIR model, the variable J(t) is the cumulative number of infections in a country’s population. This quantity cannot be observed directly, because the majority of infected individuals are never tested for the virus. The number of cases confirmed by tests is not a reliable proxy, because testing regimes change over the course of the epidemic, and vary among countries as well. The number of COVID-19 related deaths is also unreliable as a proxy, since infection death rates vary from one country to another, but as a measure of the time development of the cumulative number of infections it is probably the best statistic that is generally available. In this paper, we are interested in the overall shape of the epidemic curve, and not the time for epidemic onset or details depending on the distribution of the time between infection and death. For most countries the shape of the curves for infections and deaths are very similar in the sense that the former can be obtained by a trivial time translation and re-scaling of the latter
(14)$$J\left(t\right)\to s_{d}J\left(t+t_{d}\right),$$
where the scaling factor $s_{d}$ in Scandinavia is of the order 150 and the time lag between infection and death $t_{d}$ is about 20 days. After performing this re-scaling of the curve of observed deaths, we interpret the result as the curve of cumulative infection cases that appears in the SIR model (see Appendix A). In that model we encounter the time-dependent reproduction number R(t), which is more commonly used in epidemiology than the instantaneous relative growth rate γ(t). Consider now the the linearized SIR Equations (Equation 30) and (Equation 31) for the cumulative number of infected, J(t), and the number of active transmitters of the infection, I(t), as described in Appendix A. Introducing the reproduction number R(t)=β(t)/α, these equations take the form
(15)$$\frac{dJ}{dt}=\alpha RI,$$
(16)$$\frac{dI}{dt}=\alpha \left(R-1\right)I.$$

Here, the reproduction number is typically reduced with time due to countermeasures and the gradual removal of superspreaders from the susceptible population, while the recovery rate α remains constant. By solving Equation (Equation 16) and inserting into Equation (Equation 15), the latter can be written,
(17)$$\frac{dJ}{dt}=\alpha I_{0}R\exp\left(\int_{0}^{t}\alpha \left(R-1\right)dt^{\prime }\right).$$

Taking the logarithm of this equation and differentiating with respect to *t*, yields the differential equation
(18)$$\frac{dR}{dt}+\alpha R^{2}-\alpha R-\Gamma \left(t\right)R=0,$$
where Γ(t) is given by Equation (Equation 18). The equation can be solved numerically with the proper initial condition $R\left(0\right)=R_{0}$. In the asymptotic limit t→∞, $\Gamma \left(t\right)\to -\gamma_{\infty }$ and $R\left(t\right)\to R_{\infty }$, where
(19)$$R_{\infty }=1-\frac{\gamma_{\infty }}{\alpha }.$$

Let us assume that nobody infected has recovered at the time t=0, implying that J(0)=I(0). From Equation (Equation 15) we then find that,
(20)$$\gamma_{0}=\alpha R_{0}.$$

Thus, by using Equations (Equation 19) and (Equation 20), Equation (Equation 7) becomes
(21)$$J_{\infty }=\exp\left(\frac{R_{0}}{1-R_{\infty }}\right).$$

### 2.4. Fitting Procedure

The essential methodological idea in this work is to fit a mathematical model for the epidemic curve to an observed time series; in this case the cumulative number of COVID-19 related deaths. This curve has a sigmoid shape with near-exponential growth in the initial growth phase. In such situations a usual least-square fitting procedure will give a poor representation of the initial growth because large relative errors will give small contributions to the absolute least-square error. Much better representation is obtained by fitting the logarithm of the model to the logarithm of the data. This is what we do here, by fitting the function for lnJ(t) given by Equation (Equation 3) to the logarithm of the cumulative death per million time series starting at the first day this number exceeds one. We use the built-in fitting routine FindFit in *Mathematica* 12.0.0.0, which employs a least-square optimation to estimate the parameters $J_{\infty }$, $\gamma_{\infty }$, and $J_{0}$.

We have typically used about 100 days of the time series for this fitting, but have adjusted this manually for each country to make sure that we only include the first wave of the epidemic. We have only included countries which have clearly completed a first wave, so that we can reliably estimate the decay rate $\gamma_{\infty }$. For this reason, we have not analyzed many countries in the tropics or the southern hemisphere. The data for the 73 countries we have analyzed, and the fits to them, are shown in Figure A2, Figure A3, Figure A4, Figure A5, Figure A6 and Figure A7 in Appendix B. The fits are surprisingly good, and demonstrate the usefulness of the Gompertz model to give an analytical representation of these data.

## 3. Results

In this section we first present a detailed comparison of Sweden and Norway in order to highlight some of the useful information that can be drawn from the analytical representation of the epidemic curves through the fitted Gompertz functions. Next we estimate the Gompertz model parameters from 73 countries and classify them in terms of total death toll and ratio between rates of initial growth and later decay.

### 3.1. Epidemic Curves of Sweden and Norway

Figure 2a shows the development of cumulative deaths per million inhabitants in Norway and Sweden in a logarithmic plot, and the corresponding fitted Gompertz function. Sweden data converge to a total death toll which is more than ten times higher than that of Norway. The relative growth rates $\gamma \left(t\right)=d_{t}\ln J\left(t\right)$ given as the slope of each curve are plotted in Figure 2b. Equation (Equation 6) shows that these growth rates decay exponentially from an initial value $\gamma_{0}$ towards zero at a rate $\gamma_{\infty }$. Observe that the gap between Swedish and Norwegian death numbers continues to grow throughout April and May, consistent with the higher Swedish growth rate throughout this period.

Figure 3a shows the derivative $d_{t}J\left(t\right)$ for Sweden an Norway in logarithmic plot. The growth rate Γ(t) of this derivative is the slope of these curves, and is plotted in Figure 3b. It is given by Equation (Equation 8) and decays exponentially from $\Gamma_{0}=\gamma_{0}-\gamma_{\infty }$ to $-\gamma_{\infty }$. As shown by Equation (Equation 12) the total death toll $J_{\infty }$ is determined by the ratio of the initial and final slope, i.e., the ratio $\Gamma_{1}/\gamma_{\infty }$. The steeper initial growth in Sweden signified through the higher $\Gamma_{1}$ obviously contributes to the total death toll, but so does also also the lower rate on the decaying slope in June and July.

Before we try to quantify these contributions, let us look at the reconstructed evolution of the reproduction number R(t) for the two countries by solving Equation (Equation 18) with Γ(t) shifted 17 days backwards in time to account for the average delay between infection and death, and $\alpha^{-1}=5$ days representing the average infectious period. The initial value $R_{0}$ is given by Equation (Equation 20). The 17-days delay is uncertain, so the shape of $R_{0}$-curves shown in Figure 4a may be more accurate than the absolute dating. The important observation is that the relatively small, but consistent, difference between the reproduction number in Sweden and Norway throughout the entire epidemic wave has been sufficient to produce a tenfold higher number of deaths in Sweden.

Figure 4b shows that the ratio of the cumulative deaths in Sweden and Norway increases strongly throughout the entire epidemic wave, which indicates that the higher R in Sweden during the decaying phase has had a strong effect on the total death toll, i.e., people have continued dying in Sweden in May and June, long after the death rate was close to zero in Norway.

Further insight into the differences of the epidemic curves between countries can be obtained by the graphics demonstrated in Figure 5. Here we have marked the point $\left(\Gamma_{1},\gamma_{\infty }\right)$ for Sweden and Norway and represented the function $\ln J_{\infty }=1+\Gamma_{1}/\gamma_{\infty }$ in a density plot. The diagonal lines represent isolines for constant $J_{\infty }$. We observe that Sweden is located on the iso-line where $J_{\infty }=520$ and Norway on the line where $J_{\infty }=49$. In addition, it is seen, as already noticed, that Sweden exhibits a considerably higher initial growth rate $\Gamma_{1}$ than Norway, and also a lower decay rate $\gamma_{\infty }$. The main value of the diagram, however, is that it allows us to explore the effect on the death toll of hypothetical action in the two countries that did not materialize.

Judging from the final death tolls, the Swedish path does not seem as a very attractive option. However, what if the countries had chosen to follow the Swedish path initially, with strong growth in the early phase and then followed the Norwegian path with fast decay in the late phase? The end state would then be the purple cross which is located on the isoline with totally 224 deaths per million. Another option would be to follow the Norwegian path with slow initial growth followed by the Swedish path with slow decay. This would lead to the blue cross which lies on the 82 deaths per million line. This suggests that reduction of the initial growth is the most effective way to bring the total death toll in the first wave down. More precisely, the lower initial growth in Norway reduced the number of deaths by a factor of six compared to Sweden, and the faster decay in the late phase by nearly another factor of two.

It should be mentioned that the hypothetical paths discussed above may not be realizable in practice since there may be dynamical constraints that do not allow $\Gamma_{1}$ and $\gamma_{\infty }$ to be varied independently of each other. However, as described in Section 3.2 by studying a large number of countries we find a wide range of possible states in the $\left(\Gamma_{1},\gamma_{\infty }\right)$-plane, and no sign of “forbidden” areas within this range.

### 3.2. Analysis of Data from 73 Countries

Figure 6 shows the same as Figure 5, where the data for Sweden and Norway has been supplemented by 71 other countries. The epidemic curves and their Gompertz fits are shown in Figure A2, Figure A3, Figure A4, Figure A5, Figure A6 and Figure A7. Figure 6 shows that the death toll $J_{\infty }$ in the first wave covers a range of more than two orders of magnitude, from just above 3 in China to more than 800 in Belgium. Ordered by increasing death toll (see Figure 7c for the full ordered list), we have identified a few countries in the legend. China, New Zealand, Slovakia, South Korea and Japan belong to the group of countries with less than 10 deaths per million. In the group with $∼10^{2}$ deaths, we have Iceland, Norway, Finland, Austria, Denmark, and Germany. Furthermore, in group with $∼10^{3}$ deaths, United States, Sweden, Spain, Italy, United Kingdom, and Belgium.

From Figure 7c one could be tempted to conclude that the epidemic curves are very similar for countries with similar death toll, but Figure 6 and Figure 7a,b show that this is not so. The magnitudes of the growth rates vary considerably among countries with similar $J_{\infty }=\exp\left(\Gamma_{1}/\gamma_{\infty }\right)$. Some countries within one group have a rapid initial rise $\Gamma_{1}$ which is compensated by a rapid fall (high rate $\gamma_{\infty }$) in the late phase, while others have a slow initial growth and a slower decay, resulting in approximately the same death toll. As mentioned in the introduction, Sweden experienced considerably slower initial growth than Spain, but also achieved much slower decay than Spain due to weak measures to contain the epidemic compared to Spain’s radical lock-down. The result is that the two countries are located very close to each other in Figure 7c, but are strongly separated along the same $J_{\infty }$ isoline in Figure 6.

Mathematically, Figure 6 locates each country in a 3D plot which specifies the three Gompertz parameters $\left(\Gamma_{1},\Gamma_{\infty },\ln J_{\infty }\right)$. Hence, it contains all the information contained in the Gompertz description of the epidemic curve. Figure 7 can be conceived as the projections of the points in Figure 6 onto the three axes.

## 4. Discussion

The main purpose of this paper has been to extract objective information from the epidemic curves for COVID-19 related deaths on the country level by utilizing the observational fact that the curves for the first wave are quite accurately represented by the Gompertz function for many countries of interest. The results may be subject to different interpretations and raise many questions that we cannot answer in this paper. In this section, we will limit ourselves to present some subjective views on why the Gompertz model seems to fit these data so well and some viewpoints on the information presented in Figure 6 and Figure 7.

### 4.1. How to Interpret the Gompertz Model

By taking the time derivative of Equation (Equation 3), the Gompertz model can be written in the form
(22)$$\gamma \left(t\right)=\frac{d_{t}J}{J}=d_{t}\ln J\left(t\right)=\gamma_{\infty }\ln\left(\frac{J_{\infty }}{J_{0}}\right)\exp\left(-\gamma_{\infty }t\right)=\gamma_{0}\exp\left(-\gamma_{\infty }t\right),$$
which is identical to Equation (Equation 6). The relative growth rate γ represents how much one unit of *J* grows per unit time. It is closely related to the reproduction number R, which can be interpreted as the average number of new infections caused by one infected individual. If the disease transmission mechanisms remain constant over time, including the density of susceptible individuals, γ will remain constant, and the growth of J(t) will be exponential. If a large fraction of the population becomes infected, and eventually immune, herd immunity will appear as a nonlinearity in the SIR-equations (see Appendix A). This effect has not yet appeared on the country level in the COVID-19 epidemic, so changes in the transmission mechanism must cause a reduction of γ(t) over time. One effect that is seen in all epidemics even without societal action is that individuals who are particularly active in spreading the disease (so-called super-spreaders) catch the disease early and are removed from the susceptible population. As time goes, this reduces the effective reproduction number and γ. In addition, society acts in complex ways to resist the disease. If the total effect of this resistance on the rate of change of γ is proportional γ itself (which is a common property of a dissipative force), the equation for γ would take the form
(23)$$\frac{d\gamma }{dt}=-\eta \gamma ,$$
which is what we will find by taking the time derivative of Equation (Equation 22) with the resistance $\eta =\gamma_{\infty }$. Thus, we have arrived at a straightforward interpretation of the Gompertz model as a mathematical description of an unstable system where a quantity J(t) naturally grows exponentially in the absence of dissipation. The dissipative force in this system does not act on J(t) itself, but rather on its growth rate. Society does not take action in response to the death toll itself but to its growth.

### 4.2. How to Interpret Figure 6 and Figure 7

The interpretation of Figure 6 and Figure 7 is sensitive to how the sample of countries has been selected. A complete picture will emerge as more countries complete the first wave. We have excluded countries that are not well beyond the first peak in daily deaths, or countries that have entered a second wave much before the first was completed. Countries that have not reached the first peak are Argentina, India, South Africa, and Colombia, and countries that have entered an early second wave are Iran, Saudi Arabia, Iraq, and Chile.

We have also excluded countries that we strongly suspect have unreliable or irregular reporting or epidemic curves that, by visual inspection, are not well fitted by a sigmoid function. However, we emphasize that no country has been excluded from the sample based on the results of the analysis.

The relative effect of the two parameters on the total death toll is seen by traversing the sample in Figure 6 in the vertical and horizontal direction. We observe that the distribution in the $\gamma_{\infty }$-direction is much wider for small $\Gamma_{1}$ than for large $\Gamma_{1}$. In the vertical direction, $J_{\infty }$ varies two orders of magnitude among countries with $\Gamma_{1}<0.15$, i.e., there is a high variability of the decay rate $\gamma_{\infty }$ among countries with low initial growth. Some countries, like China, New Zealand, Slovakia, South Korea, and Japan, have taken strong action throughout the first wave, despite low initial growth of the death numbers. On the other hand, other countries like Brazil, have used relatively low initial growth as an excuse for non-action, resulting in a very high death toll. For countries with $\Gamma_{1}>0.15$ the range of $\gamma_{\infty }$ becomes narrower with increasing $\Gamma_{1}$. In this group, we find most Western-European countries and the United States. Finland, Norway, Iceland, Germany, Denmark, and Austria have $\Gamma_{1}$ in the lower end of this range and $J_{\infty }∼10^{2}$ or less, while for Sweden, United States, United Kingdom, Italy, Belgium, and Spain, $\Gamma_{1}$ is higher and $J_{\infty }∼10^{3}$. The overall impression is that the variability of the death toll in Western Europe and the United States is largely due to variations of the initial growth rate, and to a lesser extent due to variations of the later decay. There are exceptions, however. For instance, Sweden and Austria have almost the same $\Gamma_{1}$, but $\gamma_{\infty }$ on opposite tails of the distribution, yielding about ten times higher death toll in Sweden.

In the rest of the world, the picture is more mixed. Some low-income countries, with low initial growth rates, still end up with a high death toll because the decay rate is also low. On the other hand, high-income countries in South-East Asia suffer few deaths due to a combination of low initial growth combined with moderate or high decay rate.

The exceptionally low $\Gamma_{1}$ for China is due to the strong confinement of the epidemic to the Hubei province, which constitutes only 4.3 percent of the Chinese population. If we had treated Hubei as a country, the death toll per million would have been 23 times higher, and $\Gamma_{1}$ would have increased by a factor of ln23≈3 and become more like that of Japan. Remember that $\Gamma_{1}$ is defined as the growth rate at the time the death toll exceeds one per million, and this time comes earlier if the population is considered to be only that of Hubei. This observation underscores that geographic isolation of the epidemic to limited regions within a country is crucial in reducing the initial growth rate and the total death toll. The success of the Chinese strategy in limiting the first wave is the effective isolation of Hubei from the rest of China and the very strict lockdown within the province.

A caveat of this entire discussion is that the death rates reported from the various countries may be inaccurate. Systematic under-reporting will influence the estimate of $\Gamma_{1}$, but not that of $\gamma_{\infty }$. At present, we have not been able to make systematic corrections to these figures for all countries. Corrections based on figures for excess mortality is a possibility for some countries, but such figures do not exist for many of countries for which the official COVID-19 death rates are least reliable. Excess mortality may also give rise to under-estimation of COVID-19 deaths in countries with effective interventions, because these interventions reduce mortality from other diseases.

## 5. Conclusions

The huge variability of the death toll among countries in the first wave of the COVID-19 epidemic is puzzling. Many causal explanations have been suggested in the media, but on a world scale, little systematic work has been published in the peer-reviewed literature. The present work does not attempt a causal explanation, but seeks a quantitative characterization of the epidemic curve in terms of as few parameters as possible. We demonstrate that this is possible in terms of the Gompertz function, which describes a three-parameter sigmoid curve. One of these parameters is redundant because it only determines the time origin that determines the point in time when the death toll per million inhabitants exceeds one. The two parameters that determine the total death toll is the initial relative growth rate of the number of daily deaths and the decay rate of this number as the first wave subsides. The salient property of the Gompertz function is that the total death toll of the first wave depends exponentially on the ratio of those rates. It hence describes the exceptional sensitivity of the death toll to the values of these rates.

We illustrate these features by applying the analysis to compare the effect of initial growth rates and later decay rates of the neighbouring countries Norway and Sweden, where more than ten times higher death toll in Sweden is attributed to a higher early death rate and, to a lesser extent, a lower decay rate in the subsiding phase. It is also shown to be related to a somewhat higher reproduction number in Sweden throughout the entire wave.

The Gompertz function is shown to be well fitted to the death data for 73 countries around the world, and plots of the Gompertz parameters for these countries show that they are broadly distributed in this parameter space, suggesting that there are many routes to a given death toll. A task for future work is to find out more about commonalities between countries that are close to each other in the parameter space, and about features that distinguish countries that appear similar but end up at different locations in this space. The most straightforward approach would be some sort of multiple regression analysis by which the growth rate $\Gamma_{1}$ and the decay rate $\gamma_{\infty }$ are linked to socioeconomic variables and intervention measures.

At the time of writing, many countries are entering the second epidemic wave. It will be interesting to investigate if this wave can be analyzed by the same method and to study if parameters change from the first to the second wave. We already observe marked changes from the first to the second wave in Europe, in particular that death tolls in the second wave so far are much lower compared to the number of confirmed infections. Some of this effect may be related to more frequent testing, but the most important difference seems to be a much higher epidemic activity among young adults, who experience a very low death rate. Social interventions are also very different in the second wave, with more emphasis on testing, tracking, isolation, and quarantining of close contacts. These factors will probably make it necessary to analyze confirmed cases data, in addition to death data, in order to fully grasp the most important characteristics of the second wave. A more complete data set for the first wave, involving more countries, will also become available, facilitating a more complete classification of the epidemic curves of the COVID-19 pandemic.

## Figures and Tables

**Figure 1 ijerph-17-06487-f001:**
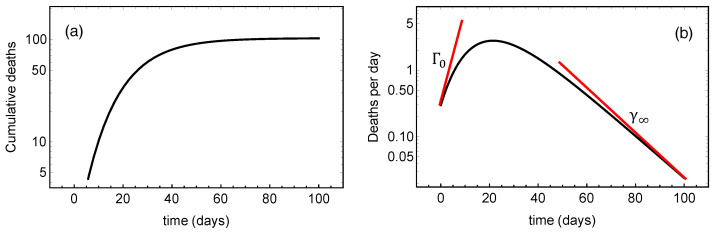
(**a**): Evolution of cumulative deaths per million J(t) in a log-plot expressed as a Gompertz function. (**b**): The evolution of the daily death number $d_{t}J\left(t\right)$ in a log-plot with the tangents at t=0 and t=∞ marked. The slopes of these tangents are $\Gamma_{0}$ and $\gamma_{\infty }$, respectively.

**Figure 2 ijerph-17-06487-f002:**
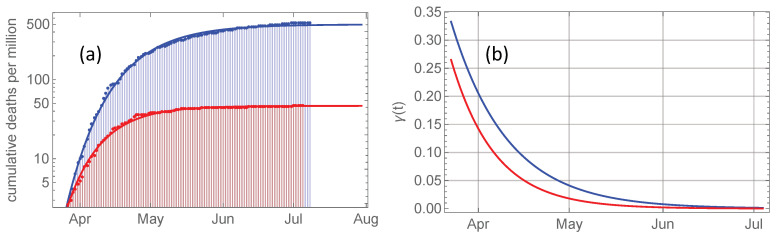
(**a**): Red bullets represent cumulative COVID-19 related deaths per million inhabitants in Norway from March 23 and onwards, and the full curve is the Gompertz curve J(t) fitted to these data. The blue bullets and curve are the corresponding for Sweden (shifted 3 days forwards). Note that the plot is logarithmic, and that cumulated death toll per million in Sweden in early July is 12 times that in Norway. (**b**): The relative growth rate $\gamma \left(t\right)=d_{t}\left(\ln J\right)$ as given by Equation (Equation 6) for Norway (red) and Sweden (blue). These growth rates are the slopes of the curves in (**a**).

**Figure 3 ijerph-17-06487-f003:**
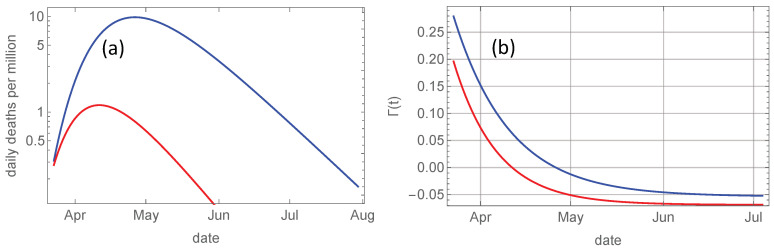
(**a**): Evolution of daily deaths per million, dJ/dt, computed numerically from Equation (Equation 12) for Norway (red) and for Sweden (blue). (**b**): The relative growth rate $\Gamma \left(t\right)=d_{t}\left(\ln d_{t}J\right)$ for Norway (red) and for Sweden (blue).

**Figure 4 ijerph-17-06487-f004:**
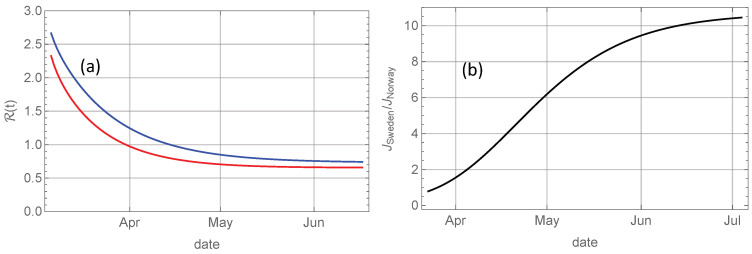
(**a**): Evolution of R(t) computed numerically from Equation (Equation 12) for Norway (red) and for Sweden (blue). (**b**): Evolution of ratio between accumulated deaths in Sweden and Norway.

**Figure 5 ijerph-17-06487-f005:**
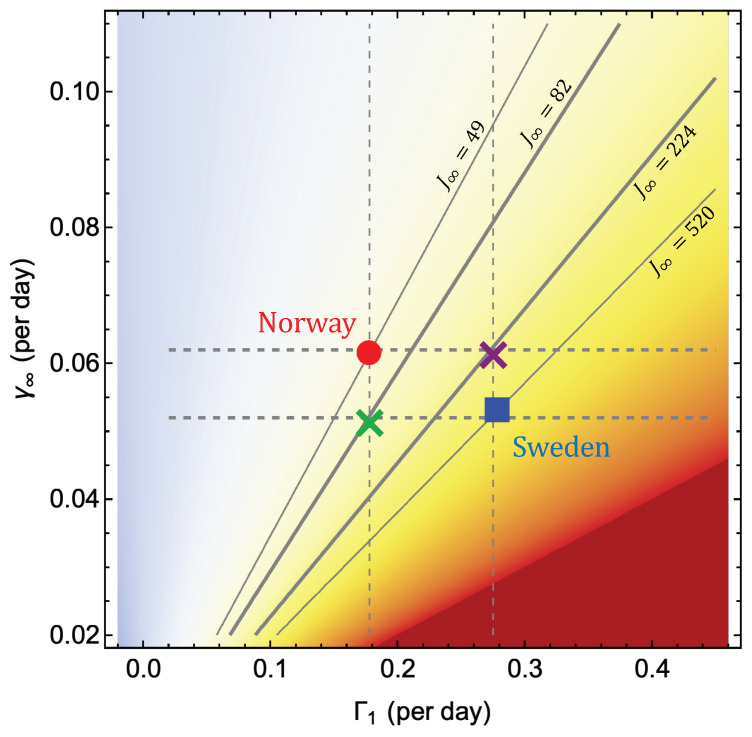
Density plot of $J_{\infty }\left(\Gamma_{1},\gamma_{\infty }\right)$ with some isolines marked, along with the points for Sweden and Norway. The crosses mark the end states for paths where the inital and final phase of the two countries are mixed.

**Figure 6 ijerph-17-06487-f006:**
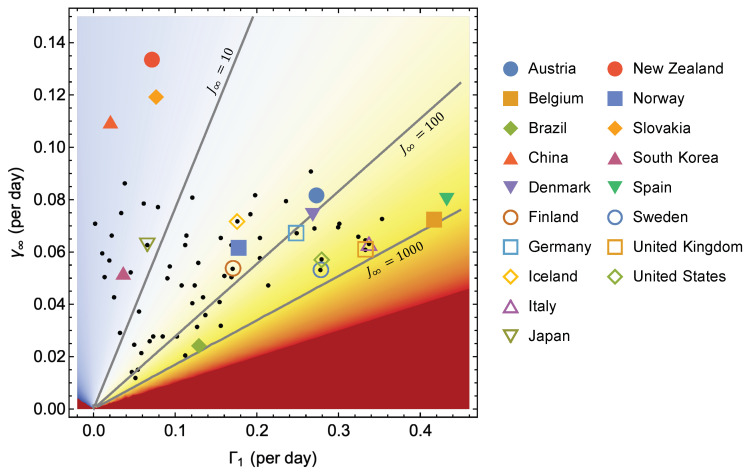
Density plot of $J_{\infty }\left(\Gamma_{1},\gamma_{\infty }\right)$ with some isolines marked, along with the points for 73 countries. The legend shows the positions for some selected countries.

**Figure 7 ijerph-17-06487-f007:**
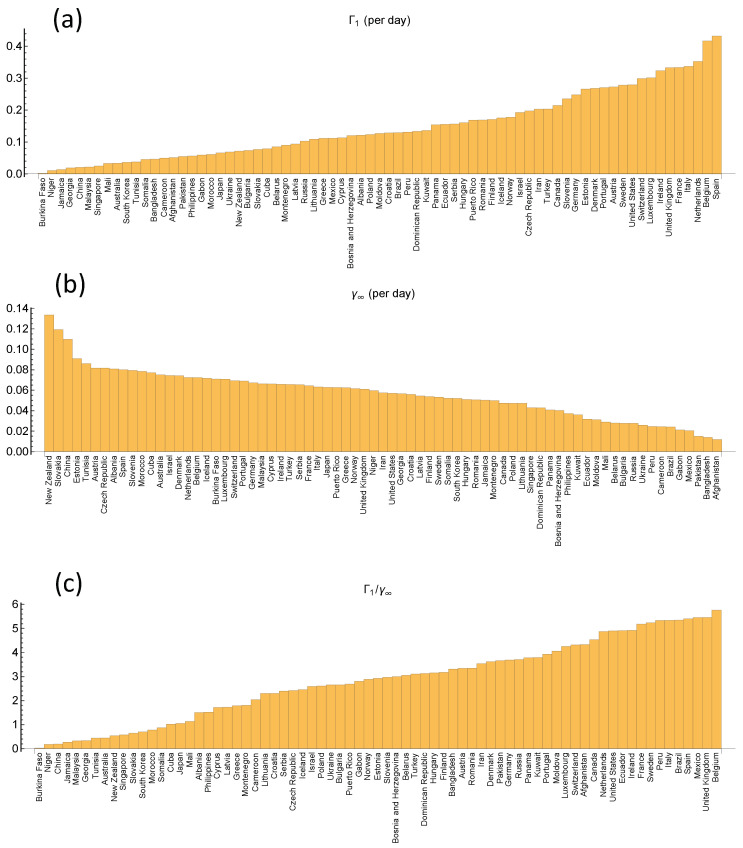
(**a**): Estimated $\Gamma_{1}$ for the 73 countries ranked from lower to higher values. (**b**): Estimated $\gamma_{\infty }$ ranked from high to low values. (**c**): Estimated $\Gamma_{1}/\Gamma_{\infty }=\ln J_{\infty }$ ranked from lower to higher values.

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
