# Peer review of "A Parsimonious Description and Cross-Country Analysis of COVID-19 Epidemic Curves"

_ijerph, 2020, doi:10.3390/ijerph17186487_

Round 1
Reviewer 1 Report
Rypdal and Rypdal showed that, for many countries, the cumulative death toll during the first wave of COVID-19 epidemic could be explained by the Gompertz function. They collected data for COVID-19 related deaths from March to July 2020 and then used the Gompertz function to explain the resulting sigmoid curves that were generated from the data. The authors mentioned the effects unreliable reporting, socioeconomic status and other important variables to consider in applying the model to study COVID-19 dynamics. The manuscript is well written and the suggestions here will contribute to better understanding of how to manage the current COVID-19 pandemic.
Minor comment
A native English speaker needs to proofread the manuscript to correct some spelling and grammar errors.
Author Response
Thanks for a positive review. We have carefully proofread the manuscript for spelling and grammar errors, and made some changes in response to the reviews. We have also corrected Eqs. (20) and (21), which contained an error (R_0-1 should be replaced by R_0) that had no consequence for the rest of the paper.
Reviewer 2 Report
A Parsimonious Description and Cross-Country Analysis of COVID-19 Epidemic Curves
This paper examines cumulative death toll of the first wave of the COVID-19 epidemic. It employs Gompertz function, which is characterized by two essential parameters, the initial growth rate and the decay rate operating as the first epidemic wave subsides. These parameters are related to socioeconomic factors and the countermeasures applied to halt the epidemic. This model implies that the total death toll growths exponentially, and so it is sensitively, on the ratio between the above rates. This comprise a criterion of distinguishing the evolution of the phenomenon in different countries and given the measure taken by them, it might be useful for the planning strategies to conform the second wave.
The work in this paper is very well performed and the presentation is very elucidating.
The first section is a good introduction to the issue explaining the need for carrying out the present work.
In the next section, the model based on Gompertz function is described in details providing the mathematical formulation and the corresponding conceptual understanding. The following sections with the procedure are also clear providing the necessary information.
The presentation of the results is also well written and understandable to the reader, so it does not leave room for clarifying questions as far as what was performed and how.
The example of Sweden and Norway can be applied to other countries given that there are data available, and the results can inform strategies for confronting a possible second wave of COVID19 pandemic.
Based on the above, I suggest publication of this paper in this form, however there is an issue, which I’ d like to see discussed in relation to the objectives of the this research:
-It is known that phenomena following a growth and decline can be described by sigmoid curves. This model is (correctly named as) a parsimonious description in the sense that two merely parameters can predict the evolution of the epidemic. The math and the model work. An issue here is related to data collection and the dependent variable examined, that is, it models the cumulative death toll. In the first wave, -it happened in all countries- the covid19 stroke people massively in the health centers (hospitals) and the deaths occurred/recorded for people having additional health problems. Thus, the deaths occurred were not due exclusively to covid19. This is why the initial growth parameter was high. The decay parameter in the present model increased sharply because of the strict direct measures imposed in a relatively controlled environment.
Coming to a possibly next wave of Covid13, this now evolves within the everyday social environment were the interactions are of different nature and the dependent variable will follow different dynamics. The question is how informative (and useful) the results of the first wave will be for predicting the progression of a potential second wave occurring within the society evolving via different interaction processes.
.
Author Response
We agree with the reviewer's comments about the different nature of the epidemic activity in the second wave that now unfolds in many countries. This was not so clear at the time of submission of this paper a month ago. We have added a discussion of this subject in the last paragraph of the paper.
Reviewer 3 Report
In the manuscript, the authors fit the Gompertz function to the cumulative death toll of the first wave of the COVID-19 epidemic in several countries. The authors discuss the characteristics of the Gompertz function and the relationship between the Gompertz model and the traditional SIR-model.
The authors carefully compare and analyze the obtained curves and Gompertz function parameters of Sweden and Norway. The analysis shows that a slightly larger reproduction number throughout the first wave of the epidemics in Sweden caused the large difference in the number of COVID-19 related deaths in these countries. The most interesting finding is that comparison of Sweden and Norway suggests that reducing the initial growth is the most effective way in pushing down the total death toll. In addition to the detailed comparison of Sweden and Norway, the authors consider several other countries as well and roughly classify the counties according to their estimated Gompertz function parameters.
The manuscript is very timely, interesting, very well written and mathematically correct. I strongly suggest accepting the manuscript for publication after a minor revision. By minor revision I here mean a very minor revision. I only have four small remarks, that are all of editorial type, that the authors could consider. These are:
- There is a small typo on page 2, line 43 (from)
- There is a small typo on page 6, line 218 (the)
- The analog to physics at the bottom of page 10 is a bit strange. You could consider dropping it.
- The names of the countries in Figure 7 are difficult to read. Is there anything that you could do to make them more readable?
I thank the authors for this manuscript. I enjoyed reading it.
Author Response
We have corrected the misprints and removed the sentences about the physics analog. We realize that the types naming countries in Fig. 7 are small, but there is so little space for them, that enlarging types makes it even more unreadable. Most readers will read the electronic version and can zoom in on it if they have problems reading on paper.